# Sleep in Residents: A Comparison between Anesthesiology and Occupational Medicine Interns

**DOI:** 10.3390/ijerph20032356

**Published:** 2023-01-28

**Authors:** Nicola Magnavita, Reparata Rosa Di Prinzio, Igor Meraglia, Maria Eugenia Vacca, Paolo Maurizio Soave, Enrico Di Stasio

**Affiliations:** 1Post-Graduate School of Occupational Health, Università Cattolica del Sacro Cuore, 00168 Rome, Italy; 2Department of Woman, Child & Public Health, Fondazione Policlinico Universitario Agostino Gemelli IRCCS, 00168 Rome, Italy; 3Department of Emergency, Anesthesiology and Resuscitation Sciences, Fondazione Policlinico Universitario Agostino Gemelli IRCCS, 00168 Rome, Italy; 4Department of Diagnostic and Laboratory Medicine, Unity of Chemistry, Biochemistry and Clinical Molecular Biology, Fondazione Policlinico Universitario Agostino Gemelli IRCCS, 00168 Rome, Italy; 5Department of Basic Biotechnological Sciences, Intensive Care and Perioperative Clinics Research, Università Cattolica del Sacro Cuore, 00168 Rome, Italy

**Keywords:** actigraph, night work, sleep quality, heart rate, sleepiness, fatigue, distress, anxiety, depression, happiness

## Abstract

Sleep deprivation is a significant risk to the health and judgment of physicians. We wanted to investigate whether anesthesiology residents (ARs) who work only one night shift per week have different physical and mental health from occupational medicine residents (OMRs) who do not work at night. A total of 21 ARs and 16 OMRs attending a university general hospital were asked to wear an actigraph to record sleep duration, heart rate and step count and to complete a questionnaire for the assessment of sleep quality, sleepiness, fatigue, occupational stress, anxiety, depression and happiness. ARs had shorter sleep duration than OMRs; on average, they slept 1 h and 20 min less (*p* < 0.001). ARs also had greater daytime sleepiness, a higher heart rate and lower happiness than OMRs. These results should be interpreted with caution given the cross-sectional nature of the study and the small sample size, but they are an incentive to promote sleep hygiene among residents.

## 1. Introduction

Sleep has a significant impact on mental and physical health [1]. Workers with sleep deprivation are exposed to disorders such as anxiety, depression [2], cognitive impairment [3] and metabolic syndrome [4,5,6]. They also have a lower resilience to stress [6] and a greater risk of occupational injuries [7] and errors [8].

Sleep problems are particularly common in healthcare workers (HCWs) [9]. Resident physicians are among the most exposed to sleep problems [10]. The sleep deprivation of resident physicians is a problem that can negatively impact their health as well as that of their patients, as sleepiness is known to increase the risk of errors [10,11].

In the past few years, several countermeasures have been taken to prevent this risk [12,13,14]. Sleep deprivation affects many categories of doctors differently, depending on the type of activity that is required at night. Anesthesiologists are known to be particularly prone to sleep problems [15] and this is especially true in conditions of understaffing and excessive workload. During the COVID-19 pandemic, resident anesthetists in a hub hospital for COVID-19 patients experienced a high rate of sleep disturbance [16,17,18,19]. The main factor in sleep loss was to be attributed both to the tension linked to the responsibility of facing a new disease with unprecedented procedures and techniques and to the excessive load of night work [19]. Mostly during the first wave, the anxiety of facing the new disease with new safety procedures and new diagnostic and therapeutic methods [20,21,22] and the fear of contracting the disease were the main reasons for sleep problems [16]. Afterwards, when the frontline staff gained full control of the new safety measures, sleep problems appeared to be mainly linked to excessive workload and lack of time for meditation and relaxation [17,18]. Since the health and alertness of anesthesiologists are critical to the quality of their assistance, the prevention of sleep problems is a priority for healthcare companies [21]. In Italy, national health authorities adopted an emergency solution to solve this problem by increasing the number of anesthesiologists, which allowed a better distribution of night work and thus reduced the workload [16,17,18,19]. According to this policy, in some European countries, such as Italy, university hospitals increased the number of available scholarships, and some residents were hired with fixed-term contracts to deal with the emergency [23]. As a result of these measures, the average number of night shifts that each resident has had to cover has decreased compared to what it was before the pandemic.

The number of night shifts covered is of particular importance in defining health risk. In Italy, in accordance with the European Directives, a “night worker” is considered to be someone who covers more than 80 night shifts a year. The discussion is open on the possibility that any health effects may also be evident for workers who work fewer than 80 nights a year. For example, does the health of resident anesthetists who do not reach 80 night shifts in a year have any differences with the health of other residents who perform the same hours of service but during the day?

Nowadays, two years after the start of the COVID-19 pandemic, the need has arisen to evaluate if the current working conditions of anesthesiology residents who work night shifts during their training may have a negative impact on their health.

We decided to carry out this study with the aims of evaluating (1) the quantity and quality of sleep in a sample of anesthesiology residents (ARs, who work night shifts) in comparison with that of occupational medicine residents (OMRs, who do not perform night work); and (2) the association between sleep and cardiac frequency, footsteps, work-related distress, fatigue, anxiety, depression and happiness.

## 2. Materials and Methods

### 2.1. Study Design and Population

This cross-sectional study was conducted between April 2022 and July 2022. Participants included 37 residents: anesthesiology residents (ARs, who work night shifts, n = 21, 11 males and 10 females, aged 29.3 ± 3.2) and occupational medicine residents (OMRs, who do not work night shifts, n = 16, 7 males and 9 females, aged 31.3 ± 2.8). Residents in Italy work the same number of hours as hospital physicians, 38 h per week [24]. Night-shift work is defined as a period of at least 3 consecutive hours in the interval between 0 a.m. and 5 a.m. [25]. ARs were working between 4 and 6 night shifts per month, so they did not reach 80 night shifts per year, which are considered in Italy the minimum level to be defined as “night workers”. All residents were asked to wear an activity tracker (see below) and complete a questionnaire with a series of validated scales. The research was conducted according to the principles of the Helsinki Convention on unpaid, healthy adult volunteers who signed an informed consent form. The research was approved by the Ethics Committee of the Catholic University of the Sacred Heart (Project 1226, 4 November 2016).

### 2.2. Activity Tracker

All residents wore an activity tracker (Fitbit^®^ Inspire HR, Fitbit Inc., San Francisco, CA, USA) on their non-dominant wrist for one week. This device objectively measured the amount of nighttime sleep (in minutes), cardiac frequency and number of footsteps per day. Data were extracted from the device, considering the mean daily values for each participant. To compare the two groups, we excluded night sleep minutes from night shifts.

### 2.3. Questionnaire

At the end of the week, each participant completed a questionnaire including sociodemographic data (e.g., age, sex) and validated scales exploring sleep (e.g., sleep quality, daytime sleepiness), fatigue, mental health (e.g., anxiety, depression), work-related distress and happiness.

Sleep quality was evaluated using the Pittsburgh Sleep Quality Index (PSQI) [26], Italian version [27], a self-report on subjective sleep quality over the previous 4 weeks with 18 questions. The first four questions were analyzed at the following times: bedtime, minutes to fall asleep, get-up time and hours of sleep per night. The next 10 questions investigated the frequency of sleeping troubles caused by different reasons (e.g., not being able to fall asleep, waking up during the night, needing to go to the bathroom, breathing problems, coughing, feeling too cold or too hot, having bad dreams and experiencing pain). The following two questions inquired about the use of sleep medication and trouble staying awake during daily activities. Each of these questions had to be answered on a 4-point scale ranging from “never” to “three times or more a week”. The last two questions asked if it had been a problem for the participant to keep up enough enthusiasm to get things done (with a 4-point scale ranging from “no problem at all” to “a very big problem”) and a subjective rating of the participants’ sleep quality on a 4-point scale from “very good” to “very bad”. The 18 items of the PSQI were summed up to a general score. Higher scores represented worse sleep quality: according to the cut-off score suggested by the authors of the questionnaire [26], subjects with scores higher than 5 were considered “poor sleepers”. In this study, Cronbach’s alpha showed good internal consistency reliability (0.792).

Sleepiness was measured with the Italian version [28] of the Epworth Sleepiness Scale (ESS) [29]. Participants rated their chances of sleeping in eight situations on a 4-point scale, scoring from 0 (“would never doze”) to 3 (“high chance of dozing”). The questions investigated the following daily activities: sitting and reading, watching TV, sitting inactive in a public place, as a passenger in a car for an hour without a break, lying down to rest in the afternoon, sitting and talking to someone, sitting quietly after lunch without alcohol and sitting in a car while stopped for a few minutes in the traffic. The results could have a minimum score of 0 and a maximum score of 24, with the normal range going between 0 and 10. A score above 10 indicated high sleepiness during the daily activities. Cronbach’s alpha for the ESS in this study showed acceptable internal consistency (0.704).

Fatigue was measured with the Fatigue Assessment Scale (FAS) [30], Italian version. The FAS consists of 10 questions, including two subscales: mental fatigue and physical fatigue. Each response was graded on a 5-point Likert scale from 1 (“never”) to 5 (“always”). Scores on questions 4 and 10 have been inversely recoded. By adding the scores of all the answers, the total FAS score was obtained, with a range of 10 to 50. A total FAS score less than 22 indicates that there is no fatigue; a score greater than or equal to 22 indicates that there is fatigue [31]. Cronbach’s alpha in this study showed acceptable reliability (0.620).

Anxiety and depression were assessed using the Italian version [32] of “Goldberg’s Anxiety and Depression Scale” (GADS) [33], referring to the previous 10-day period. GADS is composed of two scales of nine binary questions each; one point is awarded for each positive answer. A score of 5 or more on the anxiety subscale, or 2 or more on the depression subscale, indicates suspected clinically evident anxiety or depression [33]. The reliability of the GADS subscales in this study was high (Cronbach’s alpha was 0.839 for anxiety and 0.789 for depression).

The perception of work-related stress was measured using the Italian version [34] of Siegrist’s short “Effort–Reward Imbalance” (ERI) scale [35], which contains three questions for the effort variable and seven for the reward variable. All items had graded responses on a 4-point Likert scale from 1 (“totally agree”) to 4 (“totally disagree”). The resulting subscales were between 3 and 12 (effort) and between 7 and 28 (reward), respectively. The weighted effort/reward imbalance (ERI) ratio indicates the level of occupational distress. According to the literature [36], ERI values greater than 1 indicate distress [35]. Cronbach’s alphas were 0.720 for the effort subscale and 0.751 for the reward subscale.

Happiness was measured using the Ab-del-Khalek single item (“Do you feel happy in general?”) answered on an 11-point scale (0–10) [37].

### 2.4. Statistical Analyses

IBM Corp. software released in 2019 was used for statistical analysis (IBM SPSS Statistics for Windows, Version 26.0. Armonk, NY, USA: IBM Corp., release 15.0). All data were first analyzed for normality of distribution using the Kolmogorov–Smirnov test of normality.

Descriptive statistics were performed for questionnaire scores, continuous variables were expressed as mean ± SD, categorical variables were displayed as frequencies, and the appropriate parametric (Student’s t) or non-parametric (Mann–Whitney U) test was used to assess the significance of the differences between subgroups.

Correlations were calculated with the Pearson or Spearman correlation coefficients, as appropriate. Due to the small sample size, a multivariate approach to statistical analysis was not recommended; however, the effect size (according to Cohen 1988) of the r metric for the Pearson correlation coefficient was reported in order to estimate the magnitude and clinical relevance of the relationship between variables independently of the number of enrolled subjects.

The effect of sleep and stress on daytime sleepiness was studied using multiple linear regression analysis. Sleep duration, sleep quality and perceived stress were entered as predictors, and sleepiness measured with the Epworth scale was entered as a dependent variable. After adjustment for age and gender, a *p*-value <0.05 was considered statistically significant.

## 3. Results

ARs had a shorter sleep duration than OMRs, according to sleep recordings. On average, OMRs slept one hour and twenty minutes more than ARs every night (*p* < 0.001). The heart rate of ARs was also significantly higher than that of OMRs (*p* < 0.001). Conversely, the physical activity of the two groups, evaluated as the average number of footsteps taken, showed no significant difference (Table 1).

The analysis of subjective data collected through questionnaires revealed that in both groups, perceived stress was on average high, above all due to the effect of effort, which has an average score of 8.2 + 2.2, which is 68.3% of the theoretical maximum of the scale (12 points), while the rewards obtained from work have an average score of 13.7 + 3.7 (48.9% of the scale maximum). Work-related fatigue was not different in the two groups, while sleepiness was much greater in ARs than in OMRs: the difference was highly significant (*p* < 0.001). Residents who do not work night shifts also reported a higher level of happiness in life than ARs, while anxiety and depression levels were not significantly different between the two groups (Table 1).

We investigated the correlations between the variables collected in the study in the set of residents and in the two sub-groups of ARs and OMRs (Table 2). Age had no direct correlation with any of the variables, while gender was correlated with anxiety; in fact, female doctors had higher levels of anxiety than males (Table 3). Sleeping time was significantly related to happiness, whereas heart rate was negatively related to happiness. Poor sleep quality was positively associated with anxiety, depression, daytime sleepiness and fatigue and negatively associated with happiness. As expected, anxiety and depression were strongly correlated with each other and negatively correlated with happiness. Perceived fatigue appeared to be related to stress, anxiety, depression and poor-quality sleep (Table 2).

Multiple regression analysis indicated that the major determinant of daytime sleepiness in the whole group was sleep quality; sleep duration, in fact, did not show a significant predictive value in the multivariate model, and occupational stress was also not significant (Table 4). Only in the subgroup of Ars is sleep duration a predictor of sleepiness, as is age.

## 4. Discussion

This study demonstrated that resident anesthesiologists who work night shifts, even if the number of such shifts is smaller than what is considered likely to alter biorhythms, sleep significantly less than their colleagues who are not engaged in night shifts and report a higher level of daytime sleepiness. Sleep quality was also correlated with daytime sleepiness in residents. ARs also had a higher mean heart rate than OMRs, while walking the same number of footsteps in their daily routine. This is probably to be interpreted as evidence of the greater criticality of the clinical tasks required of anesthesiologists compared to those performed by OMRs. The immediate transition from rest to full attention that is typical of ARs shifts is not frequent in OMRs activities, which are generally planned in advance.

The reported levels of occupational stress, fatigue, anxiety and depression did not show differences between ARs and OMRs. However, ARs were significantly less happy than OMRs, and this finding deserves further investigation.

The results went in the direction suggested by the literature. It is not surprising that anesthesiologists sleep less than non-night-shift residents. It is well known that night workers can develop a syndrome called Shift Work Sleep–Wake Disorder (SWSWD) [38]. A meta-analysis of longitudinal studies confirmed that SWSWD is linked to a higher overall risk of negative mental health outcomes, specifically for depressive symptoms [39]. Shift work has a negative impact on cognitive functions [40] and increases the incidence of metabolic disorders [41]. However, the resident anesthesiologists recruited for this study were not night workers but covered only a limited number of night shifts and therefore should not have had any health problems. The shorter sleep duration in Ars compared to that of other residents could indicate insufficient recovery after night shifts, or it could be the effect of other factors, such as workload and responsibilities, which may interfere with sleep. The effects of workload, anxiety and emotional factors have been invoked to explain sleep disturbances reported by healthcare workers during the COVID-19 pandemic [42,43]. The female gender seems more prone to such disorders [44]. Healthcare students, on the other hand, are at risk of sleep disturbances from stressful academic and clinical workloads, even if they do not work night shifts [45]. The higher average heart rate measured in the ARs could indicate that they have a higher mental workload than the OMRs, and this may partially justify the lower sleep time.

An aspect that can significantly differentiate the work of ARs from that of OMRs is that the former have been professionally exposed to calls for emergency interventions, while the latter have not. Emergency and first aid at night have a greater mental weight than the planned prevention work carried out by the OMRs during the day. Healthcare personnel have reported hypomanic symptoms brought on by on-calls with inadequate sleep, demonstrating the impact of working conditions on their wellness and raising questions about their ability to make decisions after long work shifts [46]. According to an observational study, after completing a night call duty, anesthesiologists’ reaction times significantly rose; performance decline was linked to a larger subjective reliance on avoidance as a coping mechanism [47]. Sleep deprivation affects residents’ ability to handle crisis situations in anesthesia. The main mistakes that were made were incorrect drug administration and dosage, a failure to recognize hypotension in time and a failure to inform the surgical team of the situation [48]. A study on fatigue in emergency medicine showed that residents were significantly less alert at the completion of the night shift [49].

In terms of mental health, partial sleep deprivation impairs anesthesiologists’ overall mood state and cognitive abilities, causing them to become tenser, angrier, more exhausted, confused and irritable, and sleepier [50]. Additionally, prolonged duty shifts or quick returns to work with insufficient recovery time can disturb the circadian rhythm and cause acute and chronic sleep deprivation, which may lead to detrimental effects on chronic outcome measures (e.g., functional ability and work–life balance) [51]. Moreover, a bidirectional relationship between sleep and mood has been found, and substantial shifts in sleep timing have been shown to lead to shorter sleep and poorer mood [52]. In some studies, anesthesiologists presented a poorer quality of sleep, with excessive daytime somnolence, perceived stress and higher sedative use compared with other healthcare populations [53]. Stress brought on by inadequate sleep and overwork may be a factor in sudden spikes in blood pressure and sympathetic nervous system activity [54]. A study examining whether heart rate variability (HRV) varied amongst different physician specialties after recovery from day work and night call duty found that anesthesiologists had less dynamic HRV following day work and during night call duty, indicating higher levels of physiological stress [55]. The alterations that have been found in the literature on anesthesiologists in hospital service are in the same direction as the effects observed in this small group of resident anesthetists, who work night shifts less frequently and for a shorter period of time than doctors hired on permanent contracts.

The study presented here, which was conducted on a small group of residents, has the merit of highlighting a problem that deserves further study. If, in fact, young ARs already show a reduction in sleep and an increase in daytime sleepiness due to the few night shifts worked, it will be necessary to prepare adequate support measures. Organizational measures may help to balance the emotional involvement of residents in their work, which has been attributed to the gap between high professional demand and trainees’ lack of experience and knowledge [56]. A well-designed shift organization may affect the frequency of sleep disturbances, but the evidence about the best type of schedule is still conflicting. For example, night-float rotations, which were designed to alleviate the workload of residents on night call, induced disturbances of sleep and mood and decreased alertness in residents [57]. There is undoubtedly a need for well-designed longitudinal studies that compare the different options in order to find the organizational methods most respectful of workers’ well-being and patient safety.

The observation that night-shift healthcare professionals are at risk for sleep deprivation and/or circadian rhythm abnormalities has led many health agencies to plan health promotion interventions [58]. Programs should endeavor to take the necessary safeguards because sleep loss affects workers’ capacity for complex rational decision making. This can be accomplished by keeping nighttime complexity and decision-making speed as low as possible. Additionally, some organizational and individual actions can hasten the body’s adaptation to a new shift and lower the risk of circadian rhythm disorders.

The most Important limitation of this pilot study was the small sample size. Studies conducted on a larger number of residents may provide more reliable results. A second important limitation is the cross-sectional nature of the study, which does not allow for asserting the causality of the observed associations. Only a longitudinal study could indicate whether the lower level of happiness observed in anesthesiologists was a consequence of occupational exposure, or whether it was rather an accidental finding. Finally, the technical characteristics of the device used must also be considered. Even if a comparative study of the products available on the market indicates that the used device is reliable [59], we have observed that, since it is very sensitive, connection could be lost when detaching from the skin (e.g., during the night due to accidental movements that cannot be controlled or limited), As a result, micro-awakenings were frequently recorded and a lower number of hours of actual sleep were registered. We found that this problem occurred in approximately 10% of registrations, with equal frequency in ARs and OMRs. In two cases, the frequency of interruptions was high enough to require a repetition of the registration.

## 5. Conclusions

The observation that covering only one night shift per week is associated with a very significant reduction in sleep duration in ARs and is accompanied by increased daytime sleepiness should stimulate further longitudinal research on more numerous case series than this one pilot study. If confirmed, the results should lead to changes in work organization in order to favor the recovery of correct sleeping conditions and alertness. Mitigating measures should aim at implementing a sleep management system, i.e., a coordinated sequence of interventions at various levels (structural, organizational and individual) for the improvement of sleep hygiene and the prevention of drowsiness [60]. Healthcare companies have a vested interest in carrying out this type of intervention since daytime sleepiness is particularly hazardous for physicians performing first aid and emergency intervention in critically ill patients.

## Figures and Tables

**Table 1 ijerph-20-02356-t001:** Comparison of sleep and health-related variables between groups. For statistically significant correlations, the effect size was also reported.

Variables	ARs (n = 21)	OMRs (n = 16)	*p* Value
Sleep, minutes per night	359 ± 39	441 ± 38	0.001 L
Heart rate	80 ± 6	73 ± 5	0.001 L
Footsteps	11067 ± 2372	11784 ± 6698	0.650
Work-related distress	1.42 ± 0.45	1.68 ± 0.78	0.244
Poor sleep quality	5.33 ± 1.85	4.75 ± 3.47	0.549
Sleepiness	5.86 ± 3.31	2.94 ± 2.62	0.005 L
Fatigue	18.0 ± 4.0	17.4 ± 7.3	0.801
Anxiety	2.81 ± 2.73	2.81 ± 2.71	0.997
Depression	1.62 ± 1.40	2.13 ± 2.87	0.524
Happiness	7.19 ± 1.33	8.56 ± 1.50	0.007 L

Notes: effect size legend: L = large. For not significant comparisons the effect size span from very small to small.

**Table 2 ijerph-20-02356-t002:** Correlation between variables. Pearson coefficient. For statistically significant correlations, the effect size was also reported.

	Age	Sex	Sleepiness	Anxiety	Depression	Happiness	Poor Sleep Quality	Fatigue	Stress	Sleep Time	Heart Rate	Footsteps
Whole group												
Age	1	0.007	0.144	−0.046	−0.013	0.006	−0.001	−0.054	−0.178	0.008	−0.277	0.160
Sex	-	1	−0.054	**0.420 **** **M**	0.258	−0.102	0.258	0.189	0.044	0.276	0.150	−0.060
Sleepiness	-	-	1	0.196	0.220	**−0.416 *** **M**	**0.382 *** **M**	0.279	−0.169	−0.204	0.243	−0.138
Anxiety	-	-	-	1	**0.782 **** **L**	**−0.464 **** **M**	**0.670 **** **L**	**0.530 **** **L**	−0.300	−0.038	0.210	−0.231
Depression	-	-	-	-	1	**−0.497 **** **M**	**0.786 **** **L**	**0.768 **** **L**	−0.266	0.114	0.093	−0.243
Happiness	-	-	-	-	-	1	**−0.497 **** **M**	**−0.430 **** **M**	**0.353 *** **M**	**0.362 *** **M**	**−0.472 **** **M**	0.227
Poor Sleep	-	-	-	-	-	-	1	**0.714 **** **L**	−0.255	−0.069	0.110	−0.222
Fatigue	-	-	-	-	-	-	-	1	0.476 * M *	0.014	−0.045	−0.175
Stress	-	-	-	-	-	-	-	-	1	0.257	0.021	0.215
Sleep minutes	-	-	-	-	-	-	-	-	-	1	−0.260	0.052
Heart rate	-	-	-	-	-	-	-	-	-	-	1	−0.118
Footsteps	-	-	-	-	-	-	-	-	-	-	-	1
ARs												
Age	1	0.155	0.419	−0.073	0.092	−0.269	0.058	0.062	−0.330	−0.363	0.090	−0.034
Sex	-	1	−0.046	0.390	0.057	−0.140	0.352	0.279	−0.147	0.179	0.157	−0.180
Sleepiness	-	-	1	0.096	0.194	−0.187	0.171	0.112	−0.120	0.288	0.080	−0.086
Anxiety	-	-	-	1	**0.793 **** **L**	**−0.541 *** **L**	**0.586 **** **L**	0.409	−0.232	−0.028	0.290	−0.247
Depression	-	-	-	-	1	**−0.553 **** **L**	**0.574 **** **L**	**0.532 *** **L**	−0.319	−0.051	0.052	−0.316
Happiness	-	-	-	-	-	1	−0.332	−0.242	**0.494 * M**	0.116	−0.382	0.039
Poor sleep quality	-	-	-	-	-	-	1	0.264	0.129	0.030	0.186	−0.401
Fatigue	-	-	-	-	-	-	-	1	**−0.660 ** L**	0.099	−0.272	0.041
Stress	-	-	-	-	-	-	-	-	1	0.121	0.146	−0.287
Sleep minutes	-	-	-	-	-	-	-	-	-	1	0.064	0.066
Heart rate	-	-	-	-	-	-	-	-	-	-	1	0.105
Footsteps	-	-	-	-	-	-	-	-	-	-	.	1
OMRs												
Age	1	−0.286	0.155	−0.011	−0.168	−0.004	0.020	−0.135	−0.234	−0.306	**−0.607 * L**	0.273
Sex	-	1	0.028	0.465	0.402	−0.178	0.234	0.145	0.161	**0.503 *** **L**	0.352	−0.027
Sleepiness	-	-	1	0.430	0.462	−0.413	**0.621 *** **L**	**0.500 *** **L**	−0.067	0.037	−0.101	−0.172
Anxiety	-	-	-	1	**0.867 ** L**	−0.495	**0.794 **** **L**	**0.662 **** **L**	−0.386	−0.095	0.173	−0.267
Depression	-	-	-	-	1	**−0.696 ** L**	**0.892 **** **L**	**0.871 **** **L**	−0.294	0.104	0.327	−0.243
Happiness	-	-	-	-	-	1	**−0.635 **** **L**	**−0.617 *** **L**	0.173	−0.011	−0.232	0.317
Poor sleep quality	-	-	-	-	-	-	1	**0.892 **** **L**	−0.388	0.010	−0.038	−0.174
Fatigue	-	-	-	-	-	-	-	1	−0.401	0.054	0.089	−0.230
Stress	-	-	-	-	-	-	-	-	1	0.188	0.188	0.338
Sleep minutes	-	-	-	-	-	-	-	-	-	1	0.441	−0.042
Heart rate	-	-	-	-	-	-	-	-	-	-	1	−0.239
Footsteps	-	-	-	-	-	-	-	-	-	-	-	1

Notes: ARs: anesthesiology residents; OMRs: occupational medicine residents. ** = Correlation is significant at the 0.01 level (2-tailed). * = Correlation is significant at the 0.05 level (2-tailed). Effect size: M = medium; L = large. For not significant comparisons, the effect size spans from very small to small. Significant values are in bold type.

**Table 3 ijerph-20-02356-t003:** Gender-related differences in sleep and health-related variables.

Variables	Males	Females	*p* Value
Sleep, minutes per night	379.01 ± 47.90	409.23 ± 59.25	0.098
Heart rate	75.80 ± 6.87	77.67 ± 5.75	0.374
Physical activity, footsteps	11659.76 ± 2932.66	11108.92 ± 5943.54	0.725
Poor sleep quality	4.39 ± 2.03	5.74 ± 3.03	0.123
Sleepiness	4.78 ± 3.23	4.42 ± 3.50	0.750
Fatigue	16.67 ± 4.45	18.74 ± 6.40	0.264
Work-related distress	1.50 ± 0.46	1.56 ± 0.75	0.795
Anxiety	1.67 ± 2.20	3.89 ± 2.71	0.010
Depression	1.28 ± 1.67	2.37 ± 2.43	0.123
Happiness	7.94 ± 1.51	7.63 ± 1.61	0.546

**Table 4 ijerph-20-02356-t004:** Determinants of daytime sleepiness. Multiple linear regression.

Variables	Beta	t	*p* Value
Whole group			
Sleep, minutes per night	−0.145	−0.842	0.406
Poor sleep quality	0.403	2.335	0.026
Age	0.147	0.903	0.373
Gender	−0.119	−0.687	0.497
Work-related stress	0.002	0.014	0.989
ARs			
Sleep, minutes per night	0.587	2.917	0.011
Poor sleep quality	0.246	1.241	0.234
Age	0.652	3.125	0.007
Gender	−0.348	−1.695	0.111
Work-related stress	−0.059	−0.299	0.769
OMRs			
Sleep, minutes per night	0.142	0.530	0.607
Poor sleep quality	0.778	2.997	0.013
Age	0.187	0.766	0.461
Gender	−0.218	−0.782	0.453
Work-related stress	0.287	1.106	0.294

Notes: ARs: anesthesiology residents; OMRs: occupational medicine residents.

## Data Availability

Data are available under request to the corresponding author.

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
