# Peer review of "Sleep in Residents: A Comparison between Anesthesiology and Occupational Medicine Interns"

_ijerph, 2023, doi:10.3390/ijerph20032356_

Round 1

Reviewer 1 Report

This is a cross-sectional study to compare the sleep problems across resident anesthesiologists (ARs) and occupational medicine residents (OMRs). The methods section is fine, but some issues about the title and results should be considered.

The title is not compatible with the objective, methods and the results of the current study. So, the authors should select the better title in which the comparison of sleep problems between two groups is mentioned.

Based on the objectives of the study, it is better to put the binary group variable (ARs versus OMRs) in the model too. So, the dependent variable is sleepiness measured with the Epworth scale, and the covariates can be group binary variable (ARs versus OMRs), age, sex, sleep duration, sleep quality, and perceived stress. Therefore, the authors can determine the effect of type of residency on sleep problems independently.

The Pearson correlation was used for total data. I suggest to use this coefficient for each group (ARs and OMRs) separately.

Author Response

Dear Reviewer,

thank you for your comments and suggestions. Please find the point-by-point responses in the attached file.

Kind regards,

Reparata Rosa Di Prinzio, corresponding author

This is a cross-sectional study to compare the sleep problems across resident anesthesiologists (ARs) and occupational medicine residents (OMRs). The methods section is fine, but some issues about the title and results should be considered.

The title is not compatible with the objective, methods and the results of the current study. So, the authors should select the better title in which the comparison of sleep problems between two groups is mentioned.

R.: We sincerely thank the reviewer for appreciating our work. We completely agree with the title change, and we mentioned the group comparison in the new title: Sleep in Residents: A Comparison between Anesthesiology and Occupational Medicine Interns.

Based on the objectives of the study, it is better to put the binary group variable (ARs versus OMRs) in the model too. So, the dependent variable is sleepiness measured with the Epworth scale, and the covariates can be group binary variable (ARs versus OMRs), age, sex, sleep duration, sleep quality, and perceived stress. Therefore, the authors can determine the effect of type of residency on sleep problems independently.

R.: Welcoming the reviewer's invitation, we have added to Table 4 the results of the multiple regression conducted separately in the two sub-groups of ARs and OMRs

The Pearson correlation was used for total data. I suggest to use this coefficient for each group (ARs and OMRs) separately.

R.: We agree with the reviewer. We have added two sections in Table 2 with correlations of variables within the groups of ARs and OMRs. The study of the variables in the ARs and OMRs subgroups reported in Table 2 highlighted some erratic significances, probably deriving from the very small number of observations. Consequently, we avoided carrying out more in-depth analyzes in the two separate groups and we considered the residents as a whole in the subsequent analyses.

Reviewer 2 Report

This manuscript investigates the quality and quantity of sleep in a small sample of residents who work a single night shift per week in comparison to a sample of residents who do not work night shift. In addition, the manuscript investigates the relationship between sleep and health outcomes in these samples. Findings show those who work night shift obtain less sleep, have higher subjective sleepiness, higher heart rates, and lower happiness. Anxiety, depression, and work-related fatigue did not differ between the groups.

The introduction clearly set up the need to investigate these issues in this sample and explained how COVID-19 impacted the hospital climate. 

One concern I have is that the title does not appear to be related to the content of the manuscript. The discussion of "prevention" in the title makes me think the authors will have conducted a study on how to improve sleep problems, however, this is a correlational study looking at two different groups and assessing the relationships between sleep and health factors. I would encourage the authors to consider a title more directly relevant to what was investigated in the manuscript.

Line 192 mentions "OPs", however, I cannot find an explanation of OPs in the manuscript. 

The authors may like to consider using bold in the tables to highlight significant differences to make these more obvious to the reader.

I would be vary of concluding that the two groups undertake the same amount of physical exercise. While steps are similar, this does not mean they are otherwise similarly physically active, as steps are only one measure of physical activity.

This is an interesting and well written manuscript. The authors have done well in discussing limitations and not overstating the results in the discussion section.

Author Response

Dear Reviewer,

thank you for your comments and suggestions. Please find the point-by-point responses in the attached file.

Kind regards,

Reparata Rosa Di Prinzio, corresponding author

This manuscript investigates the quality and quantity of sleep in a small sample of residents who work a single night shift per week in comparison to a sample of residents who do not work night shift. In addition, the manuscript investigates the relationship between sleep and health outcomes in these samples. Findings show those who work night shift obtain less sleep, have higher subjective sleepiness, higher heart rates, and lower happiness. Anxiety, depression, and work-related fatigue did not differ between the groups.

The introduction clearly set up the need to investigate these issues in this sample and explained how COVID-19 impacted the hospital climate.

R.: We are very happy that the reviewer appreciated our study. The researcher's work is nourished above all by the appreciation of other researchers.

One concern I have is that the title does not appear to be related to the content of the manuscript. The discussion of "prevention" in the title makes me think the authors will have conducted a study on how to improve sleep problems, however, this is a correlational study looking at two different groups and assessing the relationships between sleep and health factors. I would encourage the authors to consider a title more directly relevant to what was investigated in the manuscript.

R.. We welcomed the suggestion and changed the title as indicated by the reviewer.

Line 192 mentions "OPs", however, I cannot find an explanation of OPs in the manuscript.

R.: Thanks for reporting a typo which we promptly corrected.

The authors may like to consider using bold in the tables to highlight significant differences to make these more obvious to the reader.

R.: We used bold font for significant values, as suggested.

I would be vary of concluding that the two groups undertake the same amount of physical exercise. While steps are similar, this does not mean they are otherwise similarly physically active, as steps are only one measure of physical activity.

R.: The reviewer is absolutely right. We carefully reviewed the text to ensure that throughout the manuscript reference was made to the number of steps and not to physical activity, which was not fully measured.

This is an interesting and well written manuscript. The authors have done well in discussing limitations and not overstating the results in the discussion section.

R.: We thank the reviewer for understanding that this pilot study has many limitations, but it could be the beginning of more in-depth research and the stimulus for preventive interventions.
